

# Boundary Layer Dynamics after Rain Fronts: High-Resolution Reconstruction and Model Validation using ground- and drone-based Measurements

Lasse Moormann[1], Friederike Fachinger[1], Frank Drewnick[1], Holger Tost[2]

[1]Multiphase Chemistry Department, Max Planck Institute for Chemistry, Mainz, 55128, Germany
[2]Institute for Atmospheric Physics, Johannes Gutenberg University, Mainz, 55128, Germany

*Correspondence to*: Holger Tost (tosth@unimainz.de)

**Abstract**

Understanding atmospheric processes enables enhancing weather forecasts and models. Research in polluted areas showed that severe rain fronts influence pollutant distribution and chemical processes in the planetary boundary layer, while studies at continental rural mid-latitude sites emphasized stratification's impact on pollutants, but neglected the influence of rain fronts. This study connects meteorological and chemical boundary layer processes during summer rain in Central and Southern rural Germany, focusing on two events: a warm front in a high-pressure system and a cold front following a convergence line. By combining near-hourly drone-based vertical profiles of the lowest 500 m, continuous ground-based observations, and ICON forecast model data, a detailed assessment of tropospheric dynamics for both events was achieved. Findings reveal that delayed nocturnal boundary layer breakup and poor vertical mixing result in weakly oxidized organic aerosol and reduced secondary aerosol formation near ground. Suppressed vertical mixing in the morning delays daytime chemical processes. A temporary reduction of $O_3$ after rain was observed, likely due to depletion from reactions with surface emissions, until mixing restored vertical homogeneity.

The ICON model accurately predicted the mixing layer height under stable conditions, but underestimated it during cold pool formation with rain showers and thunderstorms. In-situ measurements indicate that cold pool dynamics enhance subsequent convective development. These findings enhance the understanding of air mass exchange and precipitation's effects on the lower rural troposphere as well as frontal weather scenarios and atmospheric composition changes, linking local experimental and model forecast observations to larger-scale synoptic situations.

## 1 Introduction

Understanding the details of atmospheric processes during different weather events is crucial for improving weather prediction, climate modelling, and environmental risk assessments (Chen et al., 2008; Wei et al., 2011; Sziroczak et al., 2022). Fronts, particularly warm and cold fronts, play a fundamental role in shaping local and regional weather conditions in the mid latitudes,



influencing temperature gradients, wind patterns, and precipitation, as well as the chemical composition of the atmosphere.

While synoptic-scale models provide valuable insights into these phenomena, accurately capturing the fine-scale interactions within the atmospheric boundary layer remains a significant challenge (Steeneveld, 2014; Qian et al., 2016; Golzio et al., 2021; Sziroczak et al., 2022). Ground-based and aerial measurement approaches offer the potential to bridge this gap by providing spatial and temporal high-resolution data on key atmospheric variables (McWilliams et al., 2023; Moormann et al., 2025).

The formation of near-surface air mass layers is a critical factor in atmospheric dynamics, as these layers control energy

exchange, moisture fluxes, and pollutant dispersion. The structure of these layers is influenced by surface heating, turbulence, and synoptic-scale forcing, leading to variations in stability and mixing. During the night, radiative cooling at the surface leads to stable stratification referred to as nocturnal boundary layer (NBL), reducing vertical mixing and fostering the formation of temperature inversions. The evolution of the NBL during frontal passages is particularly complex, as stability transitions and wind shear interactions can enhance turbulence intermittency and alter mixing processes, which can have significant

implications for local weather phenomena, distribution of trace species, and air quality (Stull, 1988). A thorough understanding of these dynamics is essential for improving numerical weather prediction and refining models of boundary layer processes.

Measurement towers, equipped with sonic anemometers and radiometers, offer continuous, wind profiles, high-frequency turbulence, and thermodynamic data at fixed locations but are limited by their inability to capture spatial variability and typically cover a very limited vertical range (Oliveira et al., 2020). Radiosondes provide detailed vertical atmospheric profiles

deep into the stratosphere, yet they only offer snapshots or incur high costs when frequently launched (Helbig et al., 2021). The Monin–Obukhov Similarity Theory is capable of extrapolating near-surface measurements to different heights under quasi-stationary conditions, though it struggles with rapidly changing stability regimes and complex terrains (Monin and Obukhov, 1954; Markowski et al., 2019). Meanwhile, remote sensing techniques like lidar, sodar, and radar deliver high-resolution spatial and temporal profiles of wind, turbulence, and aerosol distributions without physical contact, but they are

constrained by limitations in resolving fine-scale turbulence, signal penetration issues, and reduced precision near the surface, especially under challenging atmospheric conditions (Kotthaus et al., 2023).

Recent advancements in atmospheric science have led to the integration of drone-based observations and large-scale numerical weather models to enhance the understanding of boundary layer dynamics. Drones provide flexible, spatially highly resolved measurements of temperature, humidity, and wind as well as trace gases and aerosol particles at multiple altitudes, offering a

valuable complement to ground-based instruments (Bonne et al., 2024; Radtke et al., 2024; Moormann et al., 2025). When combined with numerical weather prediction models, these measurements can improve the representation of sub-grid scale processes and help validate model simulations (Szintai et al., 2010; Zum Berge et al., 2023). However, challenges remain, including limited flight durations, regulatory constraints, and the need for robust data assimilation techniques (Elston et al., 2015; Villa et al., 2016; Moormann et al., 2025).

Previous studies have investigated frontal events with large scale model data or use large data sets of local data at ground level or measurement towers. While model data lack the high spatial resolution in the planetary boundary layer (PBL), in-situ measurements are usually limited to specific variables. Large data sets of local data cover underlying dynamics of different

rain fronts, however, they often do not cover a sufficient vertical range (Bopape et al., 2021; Helbig et al., 2021; Wang et al., 2021; Machado et al., 2024b; Machado et al., 2024a).

This study investigates and aims to reconstruct atmospheric processes during two distinct, but common weather events – a warm front in a stable high-pressure system (Sect. 3) and a cold front in a convergence zone (Sect. 4) – by integrating ground-based measurements, drone-based observations, and synoptic local-scale model data (Sect. 2). While the measurements provide a large comprehensive data set of the local meteorology as well as gas and aerosol trace matter characteristics with high temporal resolution, the model contributes the greater regional-scale picture. A detailed analysis of various variables derived

from these complementary data sources leads to a complex overview of processes, providing a more comprehensive understanding of frontal dynamics, short and longer-term impacts of rain, and the limitations of existing observational and modelling approaches.

## 2. Methodology

Data from two measurement campaigns (BISTUM23, August 2023 and BISTUM24, June 2024) were analyzed to understand

the influence of rain events on stratification and dynamical processes in the lowermost troposphere. A warm front in a stable high-pressure system (case 1, Sect. 3) and a cold front in a convergence zone (case 2, Sect. 4) were selected as examples of two kinds of rain events, which originate from different large-scale meteorological conditions and allow the analysis of various post-rain processes depending on the front type. The analysis of each case follows the same approach using a) local measurements at a ground station for continuous measurements and on-board a drone for quasi-hourly vertical profiling and

b) large-scale assessment of the meteorological conditions using a radiosonde and ICON model data.

### 2.1 Measurement sites

For the two campaigns BISTUM23 and BISTUM24, two rural sites in German low-mountain ranges were selected. BISTUM23 took place near the city of Albstadt in the Swabian Alb (48° 15' N, 8° 59' E) with the ground station at 886 m above mean sea level (a.s.l.). BISTUM24 was performed near the village of Spielberg in the Vogelsberg area with the ground

station at 391 m a.s.l. (50° 19' N, 9° 15' E). Both sites were chosen because they are in rural areas, which lowers the risk of contamination from local anthropogenic sources. The ground stations were located at the top of a hill, close to the flank of the mountain range where the aspiration usually occurs (Fig. S1). Therefore, the topographical conditions favor orographic lifting of air masses, which can facilitate deep convective events (Barros and Lettenmaier, 1994; Liu and Kirshbaum, 2025).

### 2.2 Experimental data

Measurements were performed on the same three platforms for each campaign and are listed in Table A1. Continuous ground-based measurements on-board the Mobile Laboratory (MoLa, Drewnick et al. (2012)) include the O/C ratio of the organic fraction of the submicron aerosol particles, measured with an aerosol mass spectrometer, particle size distribution (merged



data of a fast mobility particle sizer and an optical particle counter), particle number concentration (PNC), $O_3$ mixing ratio, and wind data as well as temperature, humidity, and pressure. The MoLa inlet was 6 m above ground level (a.g.l.). A ceilometer

monitored the aerosol backscatter signal above the site up to 10 km a.g.l. Additionally, during BISTUM24, 3D wind measurements at 5 m a.g.l. provided sensible heat flux ($Q_H$) and turbulent kinetic energy (*TKE*) data at 30 min-averaging intervals, as recommended for fair weather and pre-storm weather (Markowski et al., 2019).

Drone-based measurements were performed with the Flying Laboratory research drone (FLab, Moormann et al. (2025)), which provides a wide particle and gas phase dataset including $O_3$, PNC, and meteorological data. This allows the estimation of these

variables' gradients in the lowest 500 m a.g.l. as well as the calculation of the bulk Richardson number *Ri* Eq. (1) calculated from the acceleration of gravity $g$, the flight height above ground level $h_{a.g.l.}$, the virtual potential temperature $\theta v$, and the horizontal wind speeds $u$ and $v$ at ground at the respective air level:

$$Ri = \frac{g(\theta_{v,air} - \theta_{v,ground})h_{a.g.l.}}{\theta_{v,air}\left((u_{air} - u_{ground})^2 + (v_{air} - v_{ground})^2\right)} \tag{1}$$

Data from radiosondes, launched from the same site, help to verify the model data analysis of the meteorological conditions

on a larger scale and provide the convective available potential energy (*CAPE*) as an indicator of the vertical uplift forcing before and after the weather events as well as the elevated boundary layer heights (Stull, 1988).

**2.3 Model data**

Large-scale synoptical information such as 24 h-backward trajectories, the mixing layer height and radar information for the respective measurement site were derived as described in the following. The trajectories in Fig. S4 were calculated for 24 h-

backwards with the HySplit analysis tool to detect source regions outside Germany and were started from 0 m, 120 m and 500 m above ground level (Stein et al., 2015). Besides the HySplit trajectories driven by 0.25°-GFS analysis data, ICON-D2 analysis and hourly forecast data for the gaps between the analysis time events (3 hourly) were utilised to calculate corresponding mixing layer heights and backward trajectories within the German domain (Figs. S7 and S12). ICON-D2 is the operational weather forecasting model of the German Weather Service (DWD) and provides detailed weather information on

a horizontal grid width of approximately 2 km. ICON-D2 is a non-hydrostatic model, with parameterisations for shallow convection and a sophisticated boundary layer and surface exchange scheme. Even though ICON-D2 itself operates on a triangular grid, the data has been re-gridded to a regular longitude-latitude grid with a similar grid width to ICON-D2. A detailed description of the ICON model has been provided by (Zängl et al., 2015; Crueger et al., 2018) and further descriptions are available at (DWD, 2025b).

ICON-D2 uses terrain-following Gal-Chen coordinates, according to (Klemp, 2011), with 60 vertical levels in total. The lowest kilometer above ground for the Albstadt site is described with 16 vertical levels, whereas for Spielberg 15 levels cover the lowest kilometer with corresponding layer thicknesses between 30 m close to the surface and 100 m in 1 km altitude above ground. Turbulence and surface exchange processes are parameterised with a second-order scheme following (Raschendorfer, 2001).



To obtain a higher temporal resolution than the 3 hourly analysis data, forecasts for the respective two hours in between the analysis time spots have been merged with the analysis data set, i.e., forecasts with up to 2 h lead time. The forecast data was obtained from the PAMORE data archive (DWD, 2025a) for the duration of the campaign.

The backward trajectories in Figs. S7 and S12 have been calculated using a tailored trajectory program, which regrids the model data to a regular latitude-longitude-altitude grid, including the respective grid elevation of the orography. The

trajectories themselves were calculated based on a 1 min-timestep, and were finalised when the trajectory left the domain of ICON-D2 or after 24 hours.

The MLH is selected as the lowest altitude where all the following criteria in the ICON-D2 data are fulfilled: Brunt–Väisälä frequency $> 5 \times 10^{-5}$ s$^{-1}$, vertical gradient of potential virtual temperature $> 0.3$ K km$^{-1}$, and vertical gradient of absolute humidity must be curved (Wang and Wang, 2014).

Additional synoptic scale information, displayed in Figs. S2 and S3 (surface pressure, geopotential in 500 hPa), for the selected events has been obtained from ERA5 reanalysis data (Hersbach et al., 2020) as well as from radar data (WN data set) from the radar network of the DWD (e.g., Kreklow et al. (2019)). The radar and trajectory data were obtained from the DWD data server (DWD, 2025c) during the campaigns. The respective satellite images are obtained from EUMETSAT from the MSG SEVIRI (Schmetz et al., 2002) instrument. Frontal lines in Figs. S2 and S3 were manually added to highlight strong pressure gradients.

**3 Case study I: Delayed breakup of NBL during warm front rain in high-pressure system**

Our first case study focuses on the basic characterization of different air masses and investigates how strong stratification can suppress turbulence and delay the breakup of the NBL until the afternoon. This situation took place at 20 June 2024 at the Spielberg site.

**3.1 Synoptic situation and local meteorology**

The synoptic map in Fig. S2 shows that the measurement site was centrally located in a large high-pressure system that covered substantial parts of western and central Europe with an indicated warm front crossing the measurement site. Two rain events with different intensities can be attributed to the warm front: first rain: ~20 min duration, 0.5 mm accumulated rainfall starting at 11:00; second rain: ~10 min duration, 0.1 mm precipitation starting at 15:20. FLab measured three vertical profiles before and five after the major rain event (Fig. 1), allowing investigation of hour-scale influences of the rain event on the atmosphere

in the lowermost 500 m a.g.l. After the second rain event, no profiling flights were possible due to temporary airspace restrictions. Note, the following height ranges describe the height above ground level and time is given in local time (LT = UTC + 2 h), unless otherwise noted.

At 500 hPa a stable ridge was located above central Europe, leading to large-scale subsidence. Radio soundings before and after the rain event show no change in tropopause height or *CAPE* (Fig. S5), suggesting no significant change due to the front.

The high-pressure system appears to be stable, and the backward trajectories show no uplift as would be expected for warm



fronts, i.e., the uplift at the warm front is compensated or suppressed by the large-scale subsidence. However, from the ceilometer data a cloud uplift from 3 km to 5 km (or a trailing higher midlevel cloud) can be deduced immediately after the first rain (Fig. S6). The immediate but short-lived dynamical stabilization above the residual layer (RL) after the rain event and the hardly changed conditions below the RL lead to the assumption that the trajectories may not show any uplift due to a

strong RL and that the rain event can indeed be described as a warm front (see Sections 3.2 and 3.3.1). A follow-up rain event like the one at 15:20 is typical for warm fronts.

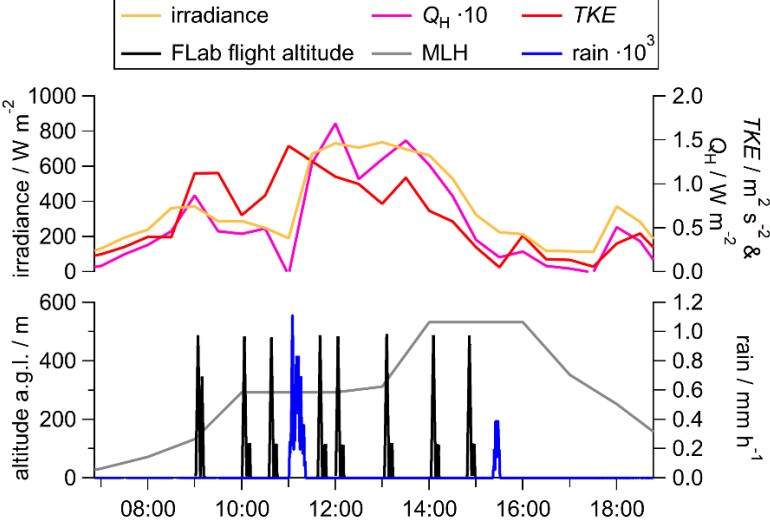

**Figure 1: The turbulence kinetic energy (*TKE*, red) increases in the morning, without being reduced prior to and during the first rain event, in contrast to the time series of irradiance (orange) and sensible heat flux ($Q_H$, pink, all 30 min averages). The strong**
**correlation between irradiance and $Q_H$ during the rain period indicates that there is no energy conservation at the ground. The hourly mixing layer height (ICON model MLH, gray) reflects stratification in 300 m a.g.l. in the morning, so FLab (drone flight altitude, black, 1 s data) measures inside and above the NBL before and directly after the main rain event (blue, 1s data).**

## 3.2 Driving forces of air mass mixing

Stratification, such as the NBL that forms near the ground during the night, usually dissipates by midday due to convective
forcing. Irradiance is the driving heat source that provides the necessary energy for the buoyancy of the near ground-level NBL. Typically, irradiance heats the ground, which conserves energy during cloud cover or darkness, and creates turbulences in the air, which release energy by dissipating eddies (Stull, 1988).

The strong correlation between irradiance and sensible heat flux throughout the day (Pearson correlation coefficient $r = 0.87$) indicates that sensible heat flux is generated mainly by direct irradiance and is not driven by stored energy from the ground
(Fig. 1). During the 11:00 rain event, irradiance and hence the sensible heat flux reach a minimum, while *TKE* is available independently – most likely due to advected air masses. The existing *TKE* and the small amount of sensible heat flux are not sufficient to elevate the NBL prior to rainfall. During the first two hours after rainfall, the thermal energy in the NBL must still accumulate before mixing of the low-level residual layer with the free tropospheric air can occur between 13:00 and 14:00, as




the model predicts. To provide evidence for this suggested development and a more holistic understanding, the air masses in
the lowest 500 m need to be characterized with in-situ data.

## 3.3 Characterization of different air masses

The air masses are characterized in reference to the 11:00 rain event primarily with FLab measurement data, which provide
information on air mass stability and history. To reduce statistical and measurement uncertainty, the presented FLab-related
data have been binned in 100 m increments for the height ranges from 0 to 500 m (see Figs. 2 and 3). Note that strong gradients
which are often measured within a few tens of meters above the ground are averaged out in the 0 to 100 m bins. To account
for concentration changes over time, the $O_3$ and PNC data measured on-board FLab were corrected for temporal trends using
the analogous continuous ground-based data measured by MoLa. The identification of different air masses (I, IIa, and IIb) and
the underlying processes discussed in the following subsections lead to a schematic description of the lowermost troposphere
after rainfall which is presented in Fig. 4.

### 3.3.1 Air mass stability

Stability indicators for vertical stratification like the bulk Richardson number $Ri$ and the gradient of the equivalent potential
temperature $d\theta_{eq}/dz$ are derived from FLab measurements. Air masses are considered statically unstable for negative $Ri$,
dynamically unstable for positive $Ri < 0.25$, and dynamically stable, i.e., laminar, for $Ri > 0.25$, whereas a negative $d\theta_{eq}/dz$
implies instability and positive $d\theta_{eq}/dz$ stable conditions (Stull, 1988). The *TKE* data, derived from the ground-based wind
measurements (Fig. 1) show consistent turbulent conditions in the lowermost 200 m in agreement with $Ri = -0.5 \pm 0.1$ and
$d\theta_{eq}/dz < 0$ for all flights after the rain event (Fig. 2). A sharp increase of the $Ri$, but still negative $Ri$ and $d\theta_{eq}/dz$ at 250 m
altitude indicates the boundary between the two stratified statical unstable layers before the rain (Fig. 2b). However, above
300 m, 15 min after the rain, air masses are suddenly stabilized, indicated by $d\theta_{eq}/dz = 38.5$ K km$^{-1}$ and contain a dynamically
stable flow (Fig. 2a). The positive $d\theta_{eq}/dz$ persists at least until completion of the flight 100 minutes after the rain event and
then becomes progressively negative, simultaneously with the $Ri$; i.e., $d\theta_{eq}/dz$ and $Ri$ drift into a more turbulent regime in the
lowermost 500 m (Fig. 2).

The upper layer can be classified as downdrafted free tropospheric air (air mass I in Fig. 4), while the lower layer is probably
a remnant of the NBL which is still present after the rain (air mass IIa in Fig. 4). The final breakup of the RL between the 3rd
and fourth flight after the rain (between 100 min and 160 min after rainfall) is induced by increasing turbulence-driven
instability above 300 m as indicated by a consistent negative $d\theta_{eq}/dz$ and a decreasing $Ri$. Probably, increased convective
forcing turns air mass IIa into air mass IIb (Fig. 4, see Sect. 3.2). Here, the continuous increase in instability with height shows
that stratification is reduced such that turbulence may increase due to larger eddies at higher altitudes.



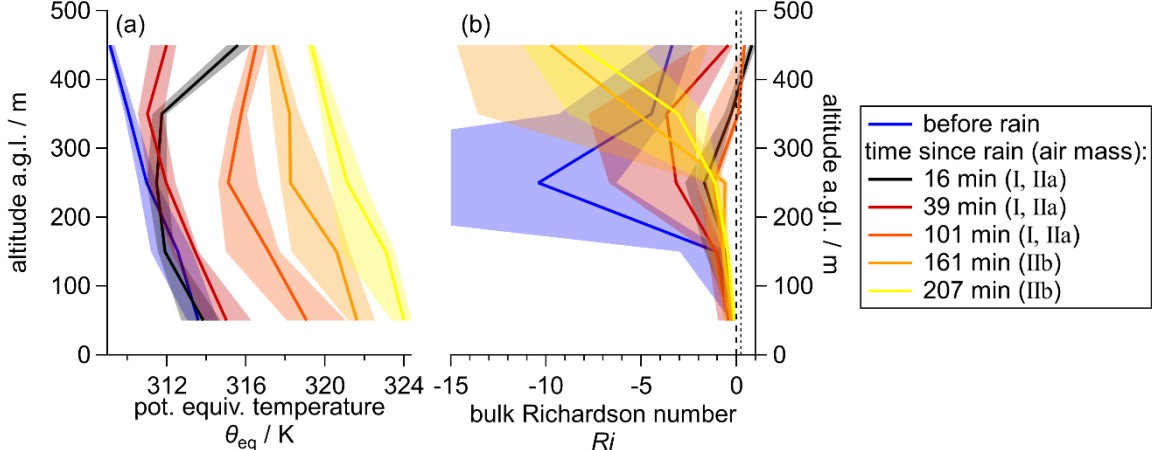

**Figure 2: The gradient of the equivalent potential temperature $\mathrm{d}\theta_{eq}/\mathrm{d}z$ (a) indicates unstable conditions throughout the day below**
**300 m, while the air mass above 300 m is stabilized after the rain at 11:00. The bulk Richardson number $Ri$ (right) is consistent with a successive destabilization after the rain and indicates a stratification between 200 m and 300 m before the rain. The solid traces show the median and the shaded area is the interquartile range of the binned data. Dashed lines indicate $Ri = 0$ and $Ri = 0.25$.**

### 3.3.2 Air mass composition and history

The air mass history like, e.g., travelled areas can be determined by 24 h-backward trajectories for different altitudes. Figure
S7 shows that the trajectories reaching the measurement site in the lowermost 100 m a.g.l. never travelled higher than 1000 m a.s.l. during the last six hours before rainfall, while trajectories arriving above 100 m a.g.l. at the measurement site show a strong downdraft of free tropospheric air close to the ground. According to Fig. 4, the new air mass (I) above 300 m is dynamically stable and differs strongly from the statically unstable air mass below 300 m before mixing (IIa and IIb in Fig. 4). In comparison to the in-situ measurements and the model MLH, backward trajectory tracks provide useful insight into air mass
movement but lack accuracy at the vertical 100 m scale.

As shown in Fig. 3, at altitudes above 300 m, $O_3$ levels are increased by up to 30%, compared to those before the rainfall, and PNC decreased by up to 70% after rain for at least 100 min. Contrary, concentrations in the lowest 300 m remain within a range of 10% for $O_3$ and 20% for PNC. High $O_3$ levels and low PNC can be attributed to a freshly ingested, clean air mass with a high-altitude origin (Neuman et al., 2012; Tsamalis et al., 2014), confirming the free tropospheric history of air mass
(I) before convective mixing. The influence of the rain on $O_3$ and PNC in the near-ground layer will be discussed in Sect. 4.3 in detail. After 100 min after the rain event no systematic gradient is observable for $O_3$ and PNC anymore. Consistent with the findings in Sect. 3.3.1, the stratification, observed directly after the rain event, has disappeared 2 h after the rain, and no significant differences of $O_3$ and PNC from the 100 m measurement increments can be observed in the lowermost 500 m, confirming a well-mixed layer. In addition to the photochemical production of $O_3$, mixing with free tropospheric air also causes
a rapid increase of $O_3$ at ground level (Neuman et al., 2012), potentially causing photochemical aging of the aerosol, as suggested by the strong correlation ($r = 0.94$) for the time series of $O_3$ levels and the O/C ratio of organic particulate aerosol





measured with the ground-based MoLa (Fig. S8). Aspiration of already-aged aerosols as a reason for enhanced O/C ratios is unlikely due to the stability of the $PM_1$ concentration measured in the afternoon. This strong correlation shows that a delayed breakup of the NBL can also lead to a delay in the onset of diurnal chemistry.

Taken these results in combination with the results shown in Fig. 2, the freshly ingested air mass can be attributed to the laminar and dynamically stable free tropospheric air mass I; initially, the lower-level statically unstable air mass IIa below 300 m does not show indication of mixing of air mass I with IIa. From 100 minutes after the rain, almost no gradients of $O_3$ and PNC are observed, indicating that the layering has dissipated and air masses I and IIa have mixed to form air mass IIb across the entire 500 m range (compare Fig. 4).


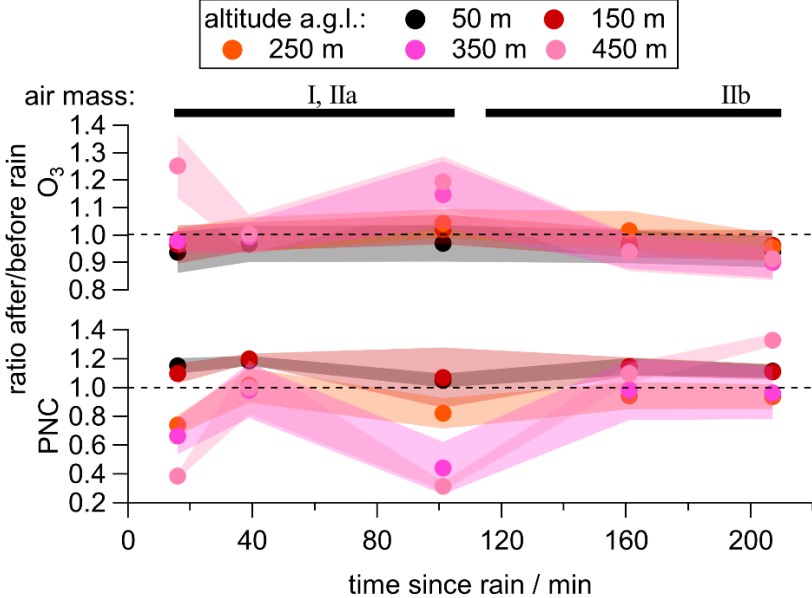

**Figure 3: The ratios of the O₃ mixing ratio (top) and the particle number concentrations PNC (bottom) after to before the rainfall (at 11:00) were calculated for different altitude increments (color-scale) for five vertical profiles measured at different times after the rain. Error bars represent the combined standard error of flights before and after the rain. Attributed air masses that are**
**identified within a flight are indicated at the top of the graph.**

### 3.3.3 Delayed NBL breakup

Boundary layer lifting can be limited or suppressed by overlying inversion layers and insufficient convective forcing. In addition to the low convective forcing due to low irradiance caused by the cloud cover, stratification is supported by the subsidence of air mass I, which is drier and warmer than the underlying NBL. Strong wind shear drives the turbulent mixing
dynamically until the temperature has increased, i.e., the energy has accumulated within the NBL and convection-driven turbulence mixes air mass I with air mass IIa 100 min after the rain. Dissipation of the boundary between the dynamically different air masses I and IIa forms the mixed air mass IIb, as shown in Fig. 4. Comparison of the observed stratification height from the in-situ data and the MLH derived from ICON data agrees within the model height resolution (± 70 m in 300 m, Fig 4).



This demonstrates that the ICON model can forecast the MLH with high accuracy at least under relatively stable large-scale

conditions.

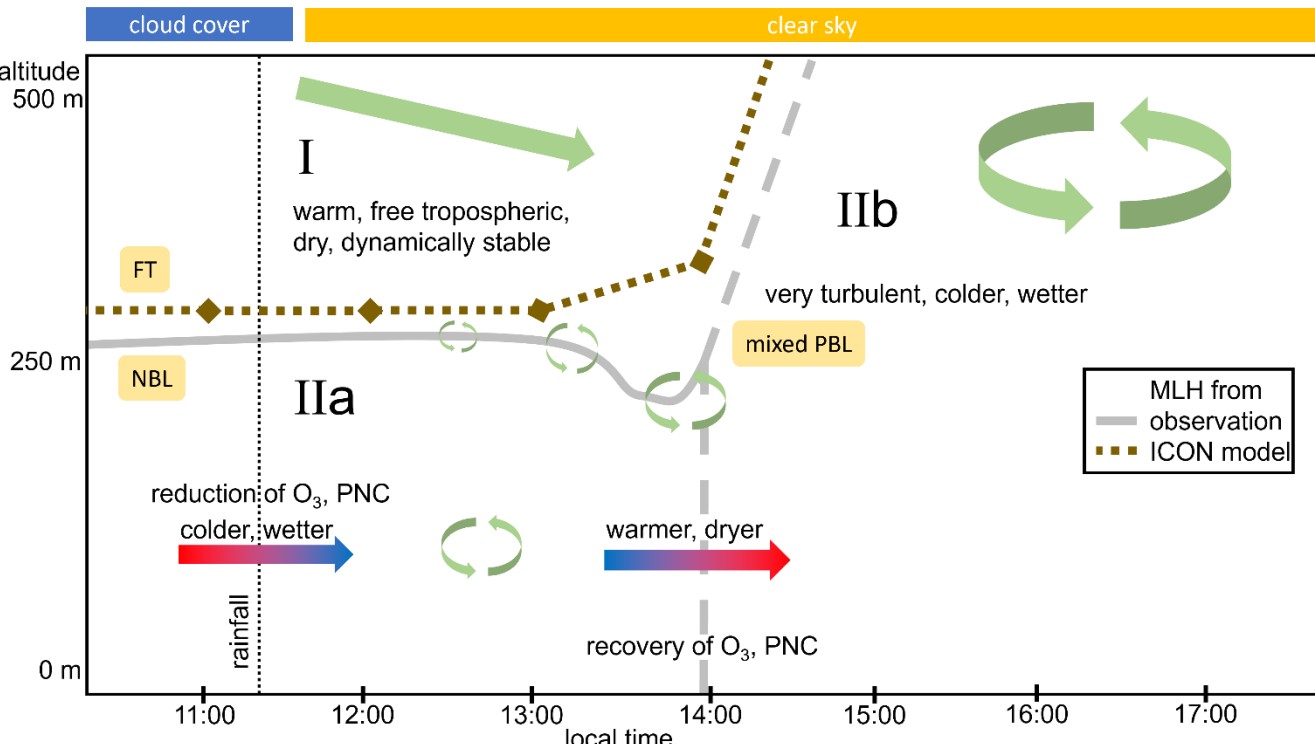

**Figure 4: A reconstruction of different characteristics and dynamics of the air masses (I, IIa and IIb, separated by grey lines) observed before and after the rainfall event. The estimated MLH from the ICON model is presented in olive with markers indicating hourly time stamps and lines between the data points to guide the eyes. Greenish arrows represent dynamic processes, multi-colored**
**arrows thermal processes.**

## 4 Case Study II: Impact of a cold front and a convergence line on the lowermost troposphere

To investigate how thunderstorms affect pollutant distribution and air mass dynamics in the lower troposphere, throughout a day when a thunderstorm occurred eleven FLab-based vertical profiling measurements were conducted hourly, as long as weather conditions allowed for safe operation.

### 4.1 Synoptic situation and local meteorology

During the 12 August 2023, four precipitation events were observed at ground level at the Albstadt site, while the ceilometer recorded seven, including three that did not reach the ground due to evaporation of rain droplets at higher altitudes (Figs. S9, S10 and Fig. 5). Even after the last rain event with a strong thunderstorm at 15:00, clouds were still present and the irradiance did not reach the levels of cloudless days, although the conditions became mild and the temperature rose by up to 10 °C.





For this day, synoptic maps show a convergence line and a cold front crossing the site from 14:40 to 15:20 (Figs. S3 and S10).
       A severe thunderstorm may have been caused by the combination of substantial lability and *CAPE* in addition to orographic
       lifting of warm air in conjunction with the convergent flow between 14:30 to 15:00. Radiosondes launched from the site
       recorded *CAPE* of 1400 J kg$^{-1}$ at 14:00, while it decreased to less than 10% of this value, i.e., 130 J kg$^{-1}$, after the thunderstorm
       (radiosonde at 16:40, Fig. S11) during mild, sunny conditions. 24 h-backward trajectories (calculated with Hysplit, Stein et al.
(2015)) indicate aspiration of air masses, amongst other across the mountain ranges of the Black Forest and the Swabian Alb,
       during the 100 km long track before they reached the measurement site (Figs. S4 and S12).

**4.2 Reconstruction of air mass exchange by cold fronts and a convergence line**

       In this subsection, similar to the approach for Fig. 4 (Sect. 3) we develop an overview schematic that describes the details of
       the boundary layer processes around the investigated event (Fig. 6). The schematic contains air masses characterized by
dynamical and chemical properties, which are tagged with roman letters (I, IIa, IIb, III, and IV) and should not be confused
       with the air masses mentioned in the previous Sect. 3.
       The meteorological situation on 12 August 2023, can be separated into a dynamically unstable pre-thunderstorm period and a
       stable post-thunderstorm period. In contrast to the situation described in Sect. 3, no ground-based flux measurement data are
       available. A consistent $Ri = 0$ at the 50 m mark during all times when the Flab was operating indicates unstable conditions,
i.e., small-scale turbulence at the ground, which are not significantly influenced by processes above 100 meters (Fig. 5a).
       Before the first rain event at 08:30, a layer boundary at the 250 m mark was identified by sign changes of the weak gradients
       for $Ri$, potential temperature, absolute humidity, $O_3$, and PNC (Fig. S13 or orange dots in Fig. 5). While positive gradients of
       potential temperature and $Ri$ indicate increasing dynamic stability with altitude, the upper air mass above 200 m is classified
       as laminar with $Ri = 0.6$ with enhanced moisture and $O_3$ levels ($O_3$ increased by 6 ppbv), and reduced particulate pollution
(PNC decreased by > 1000 particles cm$^{-3}$). Here, free tropospheric air (I in Fig. 6) overlays a weak NBL (IIa in Fig. 6) above
       200 m, similar to the case in Sect. 3.
       After the first rainfall event (0.5 mm in 15 min), the NBL remains present, reaching heights of up to 200 m as predicted by
       ICON. However, the air mass I above the NBL becomes more stable (increasing $Ri$ and d$\theta_{eq}$/d$z$) after the rainfall event. Before
       10:00, it rises to a height of 1100 m, where the ceilometer detects the upper boundary of an aerosol layer (see Fig. S10).
After a light second rainfall of 0.1 mm at 10:00, no RL and uniform gradients are observed over the whole 500 m-altitude
       range for all FLab-measured variables until 13:00. In air mass IIb mixing has eliminated the gradients. Since the first two rain
       events at 08:30 and 10:00, potential temperature near ground has increased by only 2-3 K, suggesting that diurnal heating
       primarily contributes to latent heat rather than driving vertical mixing (Fig. 5c). This observation is in agreement with very
       slow evaporation from moist soil that might explain the unusually stable equivalent potential temperature at the 50 m mark
between 10:00 and 13:00 (Fig. 5b), while the difference to the potential temperature decreases (Fig. 5c). A similar pattern is
       observed for the last rain event, although here evaporation is accelerated by strong irradiance and turbulence-driven mixing.





At 13:30, a cold front delivered 2.4 mm of rain in 10 minutes leading to a 2.2 °C decrease in air temperature and 2.1 g kg$^{-1}$ increase in humidity at ground, while equivalent potential temperature remains constant at ground, as measured with MoLa (Fig. S14). However, at 50 m altitude and above no change in temperature, humidity, and the vertical gradient of equivalent

potential temperature is observed. The slow evaporation of moisture from previous rainfall forms a localized moist patch, which, on a regional scale, might develop into a cold pool. Cold pools create a positive feedback loop: They generate broader clouds that are less affected by entrainment, leading to increased precipitation, larger moist patches, and further expansion of cold pools – ultimately fostering larger clouds with enhanced convective mass fluxes (Schlemmer and Hohenegger, 2014). Tompkins (2001) shows that along an already recovered cold pool, the increased equivalent potential temperature (here

increased by 6 K at ground and by 2.1 K at the 50 m altitude bin, Figs. 5b and S14) and water vapor trigger new convective cells, like the upcoming rain front in this case study. Due to increased latent heat flux, thermal convection is suppressed initially, leading to a homogeneously laminar air mass III with $Ri = 0.28$ above 100 m, while air in the lowest 100 m is dynamically stable due to surface roughness. The downdraft of air mass III is likely enhanced by the cold pool, as penetrative downdrafts from the mid-troposphere commonly occur after rainfall exceeding 2 mm h$^{-1}$ (Barnes and Garstang, 1982), in

agreement with model results under convergence zone conditions (Schlemmer and Hohenegger, 2014).

At 14:30 the last rain event of the day with 5.4 mm precipitation in 40 min was associated with a severe thunderstorm under a convergence line as described in Sect. 4.1. Despite the reduced $CAPE$ after the storm, the new air mass IV remains unstable. Unlike the unstable air mass III, which might be unstable due to deep-convective vertical forcing ($CAPE$), this air mass IV is likely unstable due to small-scale turbulence indicated by a suddenly negative $Ri$ above 100 m altitude and a remaining negative

$d\theta_{eq}/dz$ from 15:30 on (Fig. 5a, b). Additionally, the convection did not take place in a frontal system, thus the existing air mass was not replaced by a colder, more stable air mass. Instead, only a part of the $CAPE$ was consumed by the convective event which was initiated and substantially driven by the convergence. After the rain event, the post-convective subsidence led to cloud free conditions and thus growing instability caused by irradiance and enhanced ground temperatures (Fig. S14). Higher temperatures and thermally driven convection at the ground allow for drying of the moist near-ground layer, and for

recovery of the lowermost troposphere from the thunderstorm as thermal mixing removes the vertical gradients of potential temperature, absolute humidity, PNC and $O_3$ after the last rain event (Fig. 5 c-f). A summary of the temporal exchange of air masses and the accompanying processes is shown in Fig. 6.



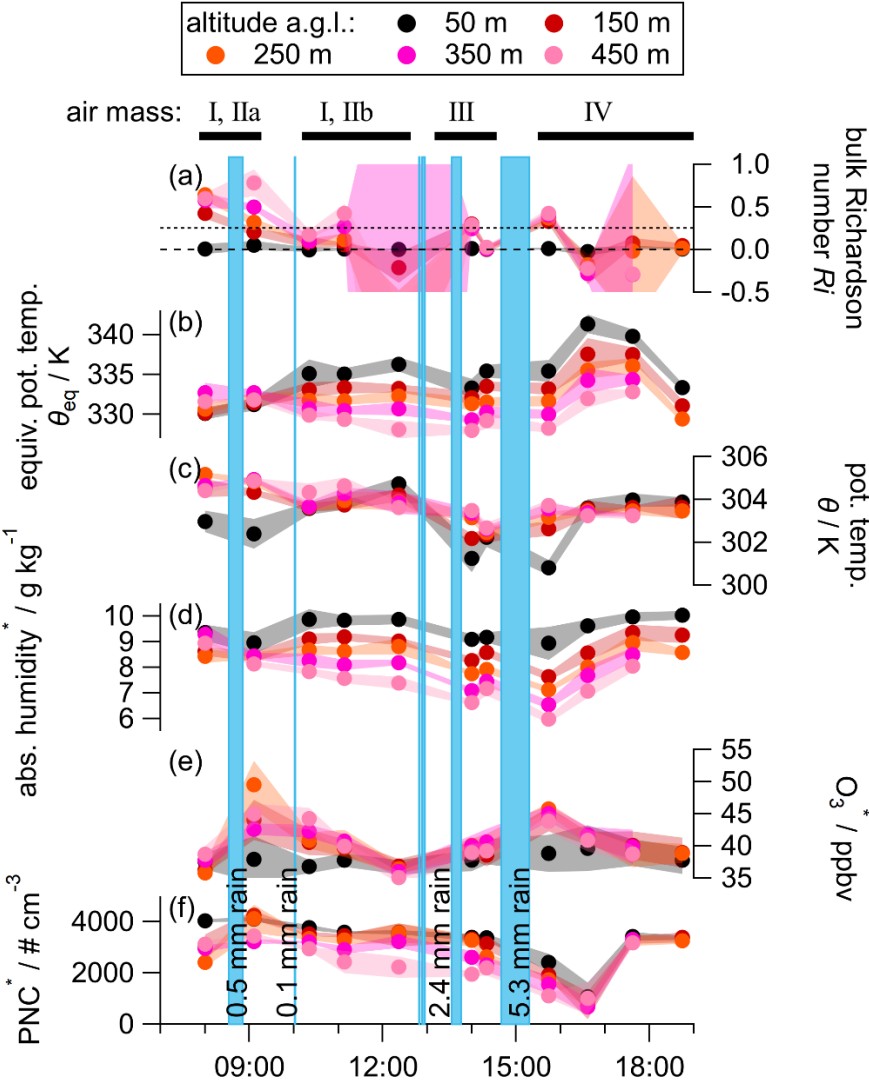

**Figure 5: Development of medians of 100 m altitude increments of the bulk Richardson number *Ri* (a), equivalent potential temperature $\theta_{eq}$ (b), potential temperature $\theta$ (c), absolute humidity (d), O₃ mixing ratio (e), and the particle number concentration (PNC, f) with time on 12 August 2023. Variables marked with * are corrected for temporal variation using the corresponding MoLa data. Error bars are derived from interquartile ranges. Dotted lines indicate *Ri* = 0 and *Ri* = 0.25. Rainfall periods are marked in blue with the amount of rain noted (precipitation at 13:00 and 14:00 were summed).**



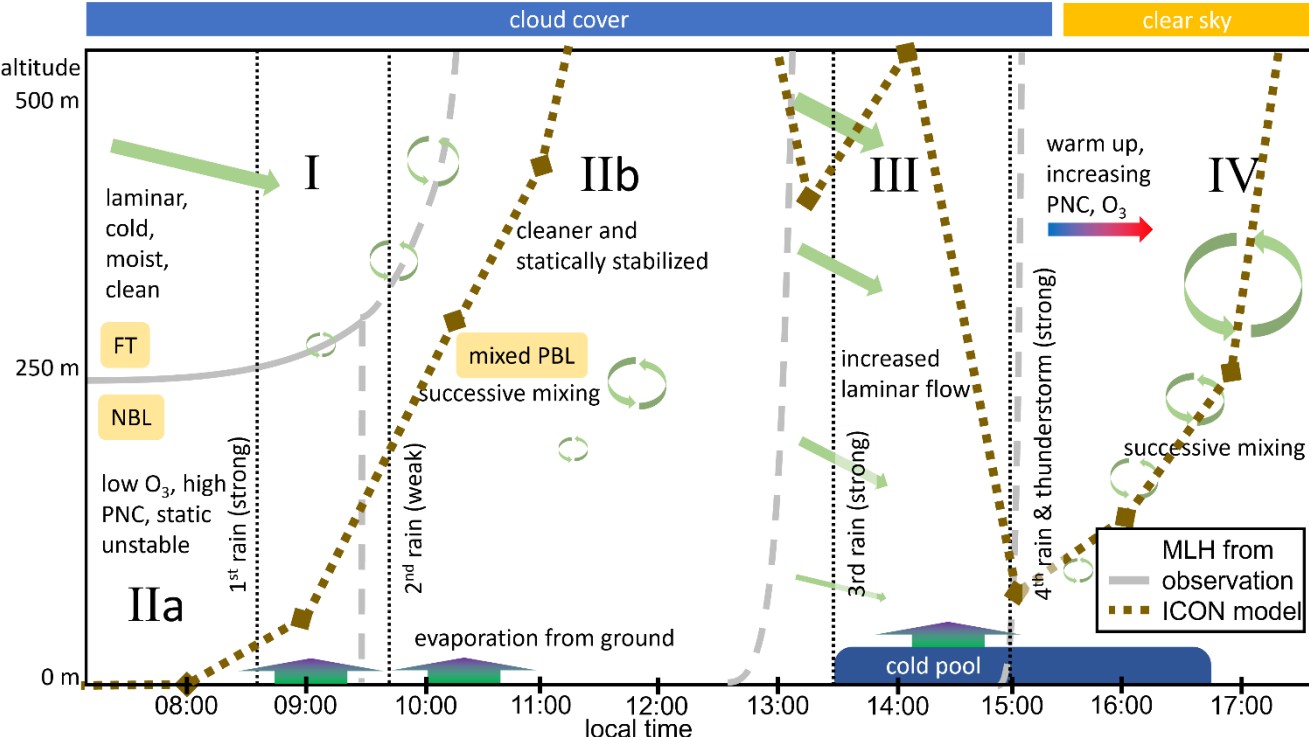

**Figure 6: A reconstruction of different characteristics and dynamics of the air masses (I, IIa, IIb, III, and IV, separated by grey lines; solid line illustrates stratification, dashed lines transition periods) on 12 August 2023. The estimated MLH from the ICON model (1-h resolution) is presented in olive with dashed lines between the data points to guide the eye. Greenish arrows display dynamic processes, multi-colored arrows thermal processes.**

**4.3 Impact of rain events on pollutant distribution**

During the day, four rain events with different amounts of precipitation occurred, leading to different changes in the trace matter distribution. Contrary to the morning precipitation events, when there was no lifting of air masses before they reached the measurement site, for the rainfall events at 13:30 and 14:30 the 24 h-backward trajectories show a lifting up to above 2000 m a.s.l. and a sudden downdraft due to the cold pool, 7 h and 1 h before arrival of the air masses at the measurement site,

respectively (Fig. S12). This downdraft dynamic was also observed by the ceilometer, which measured cloud layers subsiding from 2500 m down to 700 m a.g.l prior to the thunderstorm (Fig. S10). This downdraft results in the injection of clean, $O_3$-rich air with low particle load into the lowermost troposphere and represents the transition to air mass III (Fig. 5e, f, and Fig. 6). After each rain event, $O_3$ mixing ratios are increased at altitudes above 200 m (due to the air mass downdraft from higher altitudes, Fig. 5e), but decreased near the ground by $3.0 \pm 1.3$ ppbv, excluding the 10:00 rainfall, as shown in Fig. S15. As a

consequence, $O_3$ gradients are enhanced by 7 ppbv km$^{-1}$ for each rain event and subsequently disappear within a few hours due to convective mixing within the PBL. Reduced ground-level $O_3$ concentrations following rainfall may result from deposition or enhanced $O_3$ depletion, whereas outwash seems unlikely due to the poor solubility of $O_3$ in water. Depletion





could be driven by the increased release of primary biogenic volatile organic compounds (BVOCs) from vegetation or by peroxide-scavenging after rainfall (Bela et al., 2018; Rossabi et al., 2018; Miyama et al., 2020; Machado et al., 2024a). BVOCs have been identified as an $O_3$ sink in natural and anthropogenic conditions (Fitzky et al., 2019; Machado et al., 2024a). However, this emission-driven $O_3$ removal mechanism cannot be verified due to the lack of BVOC data.

Figure 5f highlights that for the same rain events, when $O_3$-rich air was injected into the lowest 500 m, the CPC PNC was consistently reduced by 25% (except when the NBL was still present, because air mass I is already in the free troposphere). The PNC gradient remained unchanged during and after the regeneration period, and the $PM_1$ levels measured at ground remained constant immediately after rain (Fig. S14), indicating that washout was minimal under these conditions. This is likely due to the size of the measured aerosol particles that are in the Greenfield gap and consequentially inefficiently removed by outwash (Cherrier et al., 2017).

These observations show that rain itself does not necessarily significantly influence the distribution of pollutants such as $O_3$, $PM_1$, and PNC. However, post-rain air masses determine the composition and gradients at higher levels, while rain-induced emissions from the ground may act as a sink for reactive substances as $O_3$.

**4.4 Influence of cold pool formation on model MLH**

In Section 3 we showed that the determination of the MLH using 1-h forecast data from the ICON model is feasible with high accuracy under stable conditions during a weak warm front (Fig. 4). The MLH estimation criteria in the model include the Brunt–Väisälä frequency, humidity, and potential virtual temperature gradients. Until this point of the study the MLH is used as a synonym for the height of the planetary boundary layer (PBLH), which is considered the most relevant measure separating the free troposphere and the ground-influenced layer with different dynamical characteristics and composition (Stull, 1988; Tignat-Perrier et al., 2020; Kotthaus et al., 2023). Note that other kinds of stratification occur regularly in the troposphere and are often not predictable, due to local topography, emissions, and heat reservoirs.

For the convergence line / cold front case analyzed in this section, the modelled MLH, shown in Fig. S9 and also included in Fig. 6, increases from 0 m at 08:30 to 200 m at 10:00 and 300 m at 11:00. This observation is consistent with the NBL identified from in-situ FLab data and demonstrates additionally to Fig. 4 that the MLH algorithm predicts the height of the NBL (in this case the PBL) accurately. However, between 12:00 and 18:00, the modelled MLH drops below 500 m before returning to levels above 600 m at 18:00, where it continues following its diurnal cycle (Fig. S9). The average predicted diurnal cycle of the MLH at the measurement site from Sect. 4 shows maxima of the PBL between 800 to 1000 m at 14:00 to 15:00 on clear-sky days, while being < 20 m during night (Fig. S9). The difference between the modelled MLH in the cold front case, compared to the average diurnal cycle of the PBLH, may be explained by reduced irradiance (average of 100 W m$^{-2}$ during this period). The formation of a cold pool at ground leads to an increase of latent heat flux and consequentially to less turbulent mixing. Therefore, the cold pool acts as an energy sink and suppresses energy consumption for further turbulent mixing. Although convection has driven the PBLH up to the free troposphere before the rain, the cold pool contributes to layering of the convective-driven RL above a shallow turbulent mixing layer. This stratification disappears as the cold pool dissolves.



These dynamic processes drastically alter the vertical gradients, and may have influenced the MLH model output, which relies solely on gradients. The ICON 1-h forecast models that the MLH exceeds 500 m after 11:45 and thus differs strongly from our in-situ observations. After irradiance decreased, MLH criteria were met even below 400 m due to rapidly changing gradients caused by vertical dynamics within the convergence zone. ICON data predict a MLH of 600 m at 18:00 that can be assigned

as a realistic PBLH 2.5 h after the last rain, in agreement with the mean life time of a cold pool (Tompkins, 2001). Interpolating the MLH between 12:00 and 18:00 may lead to an inaccurate estimate, because the PBLHs maximum is expected in between. In agreement, radio soundings at 14:10 and 16:40 indicate a boundary layer up to slightly above 800 hPa, i.e., 1.400 m and 1.300 m based on changing lapse rates and temperature gradients (Fig. S11, Seidel et al. (2010)). The PBLH derived from the radio soundings is 100 m to 200 m higher than the estimated average MLH for the measurement site (Fig. S9), indicating that

the PBLH has lifted due to convective forcing by the thunderstorm. A more robust parameterization of the MLH estimation for the PBLH for cold pool scenarios may consider two MLHs: a lower MLH separating an additional shallow cold pool-driven layer at ground from the previously well-mixed PBL and a second MLH separating the PBL from the free troposphere in the altitude range close to the PBL before rain events.

### 4.5 Convergence zone advection and measurement limitations

Due to unsafe flight conditions, vertical profiling during the thunderstorm was not possible; instead, ground-based MoLa data were used to analyze meteorological processes during this time (Fig. S14). Between 14:30 and 15:30, a strong cloud layer largely suppressed irradiance and minimized thermal convection, leading to dynamically deep convective inflow-driven air masses with surface wind speeds peaking at 8 m s$^{-1}$. As the rainfall starts, strong downward movement of air is implied by a descending cloud layer height, a high wind speed period, and a 40% reduction in PM$_1$ due to mixing in of unpolluted free

tropospheric air (Fig. S10, S14). Although it is known for cold pool-induced deep convective events that the horizontal wind speed is maximal before rainfall starts (Tompkins, 2001), a strong downdraft from the start of rainfall until the maximum of the precipitation rate was not described, yet, probably because observational studies describe longer-lasting rainfalls only (Young et al., 1995; Tompkins, 2001).

As the rain band advanced, heavy rain ceased and wind direction turned by 180°, an effect attributed to a moving convergence

zone (Crook and Klemp, 2000), which changed the air track from passing a rural forest (western) to the town of Albstadt (eastern, Fig. S1a). This resulted in increased NO$_x$ and PNC with lower aerosol oxidation levels, indicated by the reduced particulate organics O/C ratio (Fig. S14). After rainfall, near-surface O$_3$ was reduced (possibly due to depletion by enhanced anthropogenic or natural emissions as NO$_x$ or BVOCs, see Sect. 4.3), though it slowly recovered as the wind direction turned back from east to west after 40 min. When irradiance exceeded 400 W m$^{-2}$ at 15:35, photochemical activity led to enhanced

PM$_1$ and PNC; simultaneously, temperature and absolute humidity increased as the grassland dried.

These rapid shifts in air mass history and wind dynamics behind the convergence line highlight the need of in-situ 3D turbulence observations, e.g., with wind LIDAR (Bélair et al., 2025) to fully capture local upwelling and advection processes



within the PBL; measurements at a less polluted site (independent of wind direction) would be helpful to identify compositional changes due to the thunderstorm itself or thunderstorm-related effects like BVOC emissions from the ground.

## 5 Summary

By integrating hourly in-situ vertical profiles of the lowest 500 m with continuous ground-based observations, synoptic situation, and modelled PBLH, we can thoroughly analyze dynamical mixing, stratification, and thermal and chemical processes in two typical rain front scenarios in the continental mid-latitudes. Previous studies have often used remote sensing methods and focused separately on dynamical (Ryan et al., 2000; Oliveira et al., 2020; Helbig et al., 2021; Luiz and Fiedler, 2024) or chemical processes in various terrains (Knote et al., 2015; Miyama et al., 2020).

Our highly-resolved in situ measurements showed a significant influence of the atmospheric stratification on pollutant distribution for a stable continental PBL (Platis et al., 2016; Pohorsky et al., 2025). However, the influence of rain fronts in the mid-latitudes has not been studied so far. We fill the research gap by providing a combined analysis with highly-resolved in-situ measurements in the PBL for physical and chemical processes under frontal conditions, even including a cold pool.

In the first case study, a weak warm front associated with a large-scale high-pressure system approaches the measurement site in the morning. A warm, stable, descending air mass suppresses the vertical expansion of the NBL, while shear winds suppress turbulent mixing between the free troposphere and the NBL. Precipitation further cools the surface, converting ground thermal energy into latent heat. Although irradiance gradually increases turbulent instability, it takes about 2 h after the passage of the front for the accumulated energy to lift the NBL and to form a convectively mixed boundary layer. This delayed breakup of the NBL postpones the entrainment of free tropospheric, $O_3$-rich air and the cloud cover delays the onset of daytime photochemistry into the afternoon. The ICON forecast model accurately estimates the MLH under these conditions.

In contrast, the second case study examines a synoptic cold front trailing behind a convergence line, accompanied by several rain showers and a severe summertime thunderstorm. The ICON model underestimates the MLH after the passage of the convergence line during these unstable conditions. In order to understand the reason for the failure of MLH determination and improve weather prediction models, it is essential to reconstruct air mass exchange in the lowermost troposphere and to identify the tropospheric processes. During this cold front event, five air masses are identified, characterized by their dynamical, thermodynamical, and chemical properties. These observations reveal that a moist patch forms at ground after the morning rain; as it dries slowly, it promotes the development of a deep convective thunderstorm and triggers a positive feedback loop that leads to the formation of a cold pool. The recovery of vertical gradients and the dissipation of the cold pool in the lowest 500 m occurs over approximately 2.5 hours after the rain event, consistent with recovery processes observed in the tropics (Tompkins, 2001).

After precipitation events with more than 0.5 mm rain, we observed a decrease of ground-level $O_3$ mixing ratios, possibly due to depletion by fresh biogenic emissions; additionally, unpolluted, $O_3$-rich free tropospheric air is injected into the lowest 500



m. The gradients resulting from evaporation, $O_3$ depletion, and injection of free tropospheric air disappear within two hours after rain, driven by thermal convective mixing as temperatures and irradiance increase.

In-situ analysis and reconstructions of the lowermost troposphere reveal that air mass exchange, turbulent mixing, irradiance, and physicochemical processes each play distinct roles in defining air mass characteristics. The overview figures (Figs. 4 and 6) illustrate the post-frontal regeneration processes. These processes occur not only in rural rain events but also in other frontal situations under common meteorological conditions in the mid-latitudes. Thus, these results have broader implications on how to consider atmospheric chemical processes and compositional change in the PBL in pollutant modeling associated with rain fronts (and cold pools) due to dynamical changes. Despite being limited to two cases with hourly resolution and no in-rain flights, the study shows that combined model- and ground-based analysis can conceptually identify complex boundary layer phenomena like pollutant sinks and local stratification. This emphasizes that high-resolution vertical data are crucial for assessing individual weather events, avoid misinterpretation, and enhance forecasting and model calculations.

**Acknowledgements** The authors thank Thomas Böttger and Philipp Schuhmann (both Max Planck Institute for Chemistry) for support during the BISTUM23 and BISTUM24 campaign and Luis Valero (Johannes Gutenberg-University, Mainz) for providing the skew-*T* diagrams and corresponding *CAPE* values from radiosonde measurements.

**Code availability** The code for the FLab data acquisition and data monitoring software is available from the authors upon request.

**Author contributions** LM performed drone-based measurements, analyzed the data and drafted the manuscript. FF and FD performed ground-based measurements and related data post-processing. HT provided synoptic maps and mixing layer height calculations. All authors discussed the data and the presented results. All co-authors commented on the manuscript.

**Competing interests** The authors declare that they have no conflict of interest.

**Financial support** This work was supported by internal funding from the Max Planck Society. LM is funded by the Deutsche Forschungsgemeinschaft (DFG, German Research Foundation) – TRR 301 "TPChange" (Project-ID 428312742); similarly, HT acknowledges funding from the same project ID (subproject Z03).

**Appendix A: Instruments used for characterization of the lower troposphere**





**Table A1: Instruments used for characterization of the lower troposphere.**

| Instrument | Measured variables | Time resolution |
|---|---|---|
| **MoLa (Drewnick et al., 2012)** | | |
| **AMS[a]** | Chemical composition of non-refractory particles < 1 μm | 30 s |
| **Ceilometer[b]** | Altitude-dependent backscatter signal intensity | 30 s |
| **CPC[c]** | Particle number concentration | 1 s |
| **EDM[d]** | Particulate matter $PM_1$ | 6 s |
| **FMPS[e]** | Particle size distribution based on electrical mobility diameter | 1 s |
| **Meteorological station[f]** | Wind direction, wind speed, relative humidity, temperature, rain intensity, pressure | 1 s |
| **OPC[g]** | Particle size distribution based on optical diameter | 6 s |
| **O₃-monitor[h]** | Mixing ratio of $O_3$ | 2 s |
| **Pyranometer[i]** | Solar irradiance | 1s |

[a]HR-ToF-AMS, Aerosol Mass Spectrometer, Aerodyne Res., Inc., Billerica, Massachusetts, USA. [b]CHM 15k, Lufft Mess- und Regeltechnik GmbH, Fellbach, Germany. [c]Condensation Particle Counter Model 3786, TSI, Inc., Shoreview, Minnesota, USA. [d]Environmental Dust Monitor EDM180, Grimm Aerosol Technik GmbH, Ainring, Germany. [e]Fast Mobility Particle Sizer Model 3091, TSI, Inc., Shoreview, Minnesota, USA. [f]WXT520, Vaisala Oyj, Vantaa, Finland. [g]Optical Particle Counter Model 1.109, Grimm Aerosol Technik GmbH, Ainring, Germany. [h]Monitor 205 Dual Beam Ozone Monitor, 2B Technologies, Inc., Boulder, Colorado, USA. [i]CMP3 Pyranometer Sensor, Campbell Scientific, Inc., Logan, Utah, UK.

| Instrument | Measured variables | Time resolution |
|---|---|---|
| **Flux tower** | | |
| **Ultra-sonic anemometer[j]** | 3D wind direction, humidity, temperature | 0.05 s |

[j]CSAT3B 3-D Sonic Anemometer, Campbell Scientific, Inc., Logan, Utah, USA.

| Instrument | Measured variables | Time resolution |
|---|---|---|
| **FLab (Moormann et al., 2025)** | | |
| **Anemometer[k]** | Horizontal wind speed and direction; temperature; relative humidity; pressure | 1 s |
| **CPC[l]** | Particle number concentration | 1 s |
| **OPC[m]** | Particle size distribution based on optical diameter; temperature; relative humidity | 1 s |
| **O₃-monitor[n]** | Mixing ratio of $O_3$ | 2 s |
| **drone: Matrice 600[o]** | 3D orientation; 3D flight velocity; GPS position; wind speed and direction; altitude based on pressure level and GPS; propeller rotation rate; various internal data | ≤1 s |

[k]TriSonica™ Mini, Anemoment LLC, Lincoln, Nebraska, USA. [l]Condensation Particle Counter Model 3007, TSI, Inc., Shoreview, Minnesota, USA. [m]OPC-N3, Alphasense AMETEK®, Great Notley, United Kingdom. [n]Model 205 Dual Beam Ozone Monitor, 2B Technologies, Inc., Boulder, Colorado, USA. [o]Matrice 600, SZ DJI Technology Co., Ltd., Shenzhen, China.

| Instrument | Measured variables | Time resolution |
|---|---|---|
| **Radiosonde** | | |
| **Radiosonde[p]** | Wind speed and direction; temperature; relative humidity; pressure | 1 s |

[p]Radiosonde RS41-SGP, Vaisala Oyj, Vantaa, Finland.



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
