# Peer review of "Boundary Layer Dynamics after Rain Fronts: High-Resolution Reconstruction and Model Validation using ground- and drone-based Measurements"

_EGUsphere, 2025_

## Referee Comment (RC1)

Boundary Layer Dynamics after Rain Fronts: High-Resolution Reconstruction and Model Validation using ground- and drone-based Measurements

Lasse Moormann, Friederike Fachinger, Frank Drewnick, Holger Tost

The authors wrote a case-study based paper about how the dynamics and chemical composition of the boundary-layer changes after frontal passages. They evaluate a warm front and cold front using observations (e.g. surface-based and drone) and numerical weather prediction models. The paper itself was clearly written, and the topic is relevant as I am not aware of many studies that look to the ABL during periods of precipitation. I appreciate how the authors incorporated both observational and model data. That being said, I have a number of suggestions and comments that I think should be addressed before publication. Most importantly, I think the evidence that supports important conclusions from this manuscript are not clearly evident to me. It is likely a combination of the figures selected and the order of the text.

**Major Comments:**

- 1. There is too much material in the supplemental material. In my opinion, some of figures that are in the supplemental material are essential for supporting the conclusions that are drawn from the paper. The manuscript relies heavily on them. Those figures either should be included in the paper, or the results section should be rewritten so that the supplemental figures are not relied on as heavily as figures in the paper.
- 2. ABL Terminology and Methodology.
  - a. Lines 130-135: , how do you calculation ABL height from the observations and the model? Are they the same method? Explain this more, and discuss (somewhere) the effect of the definition of ABL height on your results. For example, ABL height very much depends on the method (e.g. Seidel et al, 2010).
    - a. There are some citations (or justification) missing here. Where do these thresholds come from? Why would Brunt-Vassilia frequency be used for mixing layer height?
    - b. How do you compute it from observations?
  - b. From reading the manuscript, I get the impression that the terms NBL, MHL and PBL are used somewhat interchangeably. They are all different concepts, and in the case of MHL and NBL, they cannot co-exist. A mixing layer implies a convective ABL and a nocturnal ABL typically wouldn't have a mixed layer to my knowledge. Please revise the terminology accordingly.
- 3. In the results, the evidence for some of the claims/conclusions is not clear. In particular, with regards to the presence of a cold pool in Case II and the presence of an NBL.
  - a. Where do you see evidence of a cold pool? If it is from ICON, I suggest including a figure that shows it. Right now, it reads as a cold pool might be there because of surface evaporation (which wasn't measured I think?) would increase after a rain event. If all the evidence is from the celimometer (Fig. S10) that should be moved into the main body.
  - b. From your results, it is not clear to me why you're considering the ABL before the rain an NBL. For example, in Case I, you have positive sensible heat flux, relatively

- strong TKE (Fig. 1) and negative Ri and negative slope in  $\frac{d\theta_{eq}}{dz}$  (Fig. 2). Same with Case II, it looks like a typical morning ABL growth (e.g. 50 m is unstable before the rain).
- c. I like Figs. 4 and 6 for making the concept easy to understand, but I don't think the data that underlies them is clear in the manuscript.
- 4. It is quite a board paper. Topic-wise, you cover ABL dynamics and chemical composition and you use many different methods. In the abstract, introduction, and summary, clarify the intent and significance of the paper. For example, what can we apply to other sites? Is this a validation of ICON?

**Technical points:**

- 1. Line 45 It is a bit confusing why MOST is brought up here in the context of observational limitations.
- 2. It would help if you included the equations for equivalent potential temperature and vertical potential temperature as they are derived variables.
- 3. Line 87 I am not familiar with "aspiration" in this context, and I don't understand the meaning here.
- 4. The order of the supplemental material is out of order compared to how it is referenced in the paper.
- 5. The order of referencing figures in the paper is also out of order (e.g. Fig. 4 is referenced in text before Fig. 3).
- 6. Some citations are the wrong type (e.g. lines 118, 119, 120, 124 and other places). If using latex, change \citep to \cite for the proper citation type.
- 7. Line 132 you haven't defined the MLH yet.
- 8. Fig. 1. Check the units of Q\_H are identified correctly. Is it truly < 2 W m-2 all day? And what is the meaning of QH \* 10 in the legend.
- 9. Line 214 rephrase the "like, e.g." combination.
- 10. Line 271 I'm not familiar with "lability" in this context.
- 11. Line 426 don't use "significant" if you didn't do a significance test.